# Mistreatment of newborns after childbirth in health facilities in Nepal: Results from a prospective cohort observational study

Ashish K. C.[1,2]*, Md Moinuddin[3,4], Mary Kinney[5,6], Emma Sacks[7], Rejina Gurung[8], Avinash K. Sunny[8], Pratiksha Bhattarai[8], Srijana Sharma[8], Mats Målqvist[1]

1 International Maternal and Child Health, Department of Women's and Children's Health, Uppsala University, Uppsala, Sweden, 2 Society of Public Health Physicians, Kathmandu, Nepal, 3 Institute of Child Health, University College London, London, United Kingdom, 4 Maternal and Child Health Division, iccdrb, Dhaka, Bangladesh, 5 School of Public Health, Faculty of Community and Health Sciences, University of the Western Cape, Cape Town, South Africa, 6 Department of Global Health, Save the Children, Cape Town, South Africa, 7 Department of International Health, John Hopkins University, Baltimore, Maryland, United States of America, 8 Golden Community, Lalitpur, Nepal

* aaashis7@yahoo.com, ashish.k.c@kbh.uu.se

**Data Availability Statement:** All relevant data are within the manuscript and its Supporting Information files.

## Abstract

### Background

Patient experience of care reflects the quality of health care in health facilities. While there are multiple studies documenting abuse and disrespect to women during childbirth, there is limited evidence on the mistreatment of newborns immediately after childbirth. This paper addresses the evidence gap by assessing the prevalence and risk factors associated with mistreatment of newborns after childbirth in Nepal, based on a large-scale observational study.

### Methods and findings

This is a prospective observational cohort study conducted over a period of 18 months in 4 public referral hospitals in Nepal. All newborns born at the facilities during the study period, who breathed spontaneously and were observed, were included. A set of indicators to measure mistreatment for newborns was analysed. Principal component analysis was used to construct a single newborn mistreatment index. Uni-variate, multi-variate, and multi-level analysis was done to measure the association between the newborn mistreatment index and demographic, obstetric, and neonatal characteristics. A total of 31,804 births of newborns who spontaneously breathed were included. Among the included newborns, 63.0% (95% CI, 62.5–63.5) received medical interventions without taking consent from the parents, 25.0% (95% CI, 24.5–25.5) were not treated with kindness and respect (roughly handled), and 21.4% (95% CI, 20.9–21.8) of them were suctioned with no medical need. Among the newborns, 71.7% (95% CI, 71.2–72.3) had the cord clamped within 1 minute and 77.6% (95% CI, 77.1–78.1) were not breast fed within 1 hour of birth. Only 3.5% (95% CI, 3.2–3.8) were kept in skin to skin contact in the delivery room after birth. The mistreatment index showed maximum variation in mistreatment among those infants born to women of relatively

**Funding:** The main study was funded by the Swedish Research Council (VR), the Laerdal Foundation for Acute Medicine, Norway, and Einhorn Family Foundation, Sweden. AKC was the grantee of Swedish Research Council (VR). MM was the grantee Laerdal Foundation for Acute Medicine, Norway, and Einhorn Family Foundation, Sweden. The funders had no role in study design, data collection and analysis, decision to publish, or preparation of the manuscript.

**Competing interests:** No authors have competing interests.

**Abbreviations:** aOR, adjusted Odds Ratio; CEONC, Comprehensive Emergency Obstetrics and Neonatal Care Services; CI, confidence interval; PCA, Principal component analysis; RMC, Respectful Maternity Care; CS PRO, Census and Survey Processing System; SNCU, Special Newborn Care Unit; SPSS, Statistical Package for Social Sciences; STROBE, STrengthening the Reporting of OBservational studies in Epidemiology.

disadvantaged ethnic groups and infants born to women with 2 or previous births. After adjusting for hospital heterogeneity, infants born to women aged 30–34 years (β, -0.041; p value, 0.01) and infants born to women aged 35 years or more (β, -0.064; p value, 0.029) were less mistreated in reference to infants born to women aged 18 years or less. Infants born to women from the relatively disadvantaged (*chhetri*) ethnic groups (β, 0.077; p value, 0.000) were more likely to be mistreated than the infants born to relatively advantaged (*brahmin*) ethnic groups. Female newborns (β, 0.016; p value, 0.015) were more likely to be mistreated than male newborns.

## Conclusions

The mistreatment of spontaneously breathing newborns is high in public hospitals in Nepal. Mistreatment varied by hospital, maternal ethnicity, maternal age, and sex of the newborn. Reducing mistreatment of newborns will require interventions at policy, health system, and individual level. Further, implementation studies will be required to identify effective interventions to reduce inequity and mistreatment of newborns at birth.

## Background

In the last 20 years, there has been an extraordinary rise in the prevalence of institutional births with over two thirds of births now taking place in health facilities [1, 2]. However, evidence has shown that the quality of care for mothers and newborns is often inadequate in health institutions, and two-thirds of deaths in health institutions can be attributed to poor quality of care [3, 4], including for newborns [5].

Health care is a fundamental human right, and persistent poor quality of care in health institutions violates this right and can also impact the confidence in the health system [6]. High-quality care for women and newborns in a health institution does not only improve health outcomes but also generates trust among those seeking health services [4, 6]. This trust towards health institutions can be reinforced if women receive respectful care during interactions with health care providers, including the medically and culturally significant experience of giving birth [7, 8]. However, several studies in South Asia and Sub Saharan Africa have reported disrespectful care of mothers during childbirth and in the postnatal period [9–11]. Disrespect and abuse during this vulnerable time ranges from direct mistreatment such as physical, verbal, and sexual abuse, to sometimes more subtle challenges such as poor communication, non-consented care, and health systems failures such as lack of meeting national or international standards [12, 13]. As result of these negative experiences, some women, especially those who are most vulnerable, such as poor women and adolescents, may hesitate to return to health facilities for births or other health care needs [13].

Until recently, the respectful maternity care agenda focused primarily on the women's experience of childbirth [12, 13], but with greater attention to newborn care, the respectful care agenda began to expand [14]. The World Health Organization included newborns in their new standards and guidelines for improving the quality of maternal and newborn care around the time of birth in health care facilities; these standards include targets for evidence-based clinical care as well as experience of care [15]. In 2019, the Respectful Maternity Care Charter was reissued to include aspects of newborn rights, such as entitlements to birth registration and not being unnecessarily separated from their mothers or families [14].

The small but growing amount of evidence shows that newborns also do not always receive respectful, evidence-based care [16], and demonstrates the need for further research [17]. Developing a set of indicators to assess mistreatment for newborns and measure respectful newborn care is crucial [18, 19]. A typology for abuse and disrespect during childbirth was proposed in 2011 by Bowser and Hill [12], and additional categories of mistreatment of newborns were proposed in 2017 [19]. These include mistreatment based on background characteristics, failure to meet standards, poor rapport with between patients and providers, and health system constraints [19, 20].

To date, there have been almost no studies conducted to assess the prevalence of mistreatment of newborns in health facilities. We conducted an observational study assessing the prevalence and factors associated with mistreatment of newborns in public health facilities in Nepal.

## Methods

The study has been reported as per the checklist for STrengthening the Reporting of OBservational studies in Epidemiology (STROBE) [21].

### Design

We conducted a prospective cohort observational study in 4 public hospitals in Nepal for a period of 18 months (14 April 2017–17 October 2018). The study was nested within a larger study of 12 hospitals in Nepal that was evaluating the impact of a quality improvement package for neonatal resuscitation care on perinatal outcomes [22]. In 4 of these hospitals, observations of labour and delivery were done to measure provider performance of neonatal resuscitation and essential newborn care. The analysis of mistreatment of newborns after childbirth was not part of the original analysis; however, these observations also captured elements of mistreatment of newborns and are presented here. The protocol for the original prospective study is published elsewhere [21]. Elements of essential newborn care are measured against the "WHO standards for maternal and newborn care in health facilities" [16].

### Setting

The 4 public hospitals were in Province 1, 3, 4 and 5 of the country. All the hospitals provided spontaneous vaginal, assisted vaginal, and caesarean section delivery. All 4 hospitals have specialized sick newborn care services. Each hospital had between 6,000 and 12,000 annual deliveries. Three of the hospitals were in the flat lands and one hospital was in a mountainous region. The labour unit in each hospital was led by skilled birth attendants and had access to neonatal resuscitation services at birth. The caesarean section percentage at the hospitals was between 18–27%. The number of nurses for labour and childbirth ranged from 6 to 8 in each hospital. During the study period, the intrapartum related mortality ratio ranged from 15 to 31 per 1000 births at the 4 hospitals.

### Study participants

All vaginally-born newborns who were spontaneously breathing at birth and born to women who were admitted into the 4 hospitals for delivery were eligible to be included in the study. Women were approached at admission to the hospital for delivery. Those who consented to the study were enrolled, along with their newborns. Newborns who did not spontaneously breathe were excluded from the study, as they required different interventions and care.

 

## Data sources

A data collection system was set up at each hospital and observations were conducted by independent clinical research nurses in each hospital's admission, labour-delivery, and postnatal units. Each data collection team in each hospital had a research site coordinator on-site. Women who consented to be part of this study were tracked from admission until discharge. Consent to participate in the study was not related to consent for clinical care. Observation of labour, delivery, and immediate newborn care was assessed using an observation checklist. Socio-demographic information on women's ethnicity, literacy, age, and previous obstetric history (parity) were collected using a semi-structured questionnaire administered at the time of discharge.

## Data management

After completing observations and semi-structured interviews with the participant, the forms were reviewed by the research site coordinator, on a daily basis. Any incomplete information and discrepancies were corrected by the site coordinator based on consultations with the data collector before being indexed. The data-entry assistant reviewed the forms from each hospital, coded the open-ended questions, and provided the data to the data-entry operator for entry.

## Sample size

This is a nested study within a larger study whose aim was to evaluate a quality improvement package on perinatal care [23]. For the larger study, an estimated 80,000 women-infant pairs were required to assess the change in intrapartum related mortality [24]. No sample size calculations were required for the nested observational component.

## Mistreatment of newborns

The indicators selected to assess mistreatment of newborns were based on the WHO's 2016 "Standards for improving quality of maternal and newborn care in health facilities" quality of care statements and process of care [15]. Deviation in the process of care for newborns after childbirth as stated in the guideline was defined as mistreatment (S1 Table).

## Clinical care practices

1. Newborns who received suctioning but did not require the intervention (e.g. already spontaneously breathing)

2. Newborns who were handled roughly defined as "spontaneously breathing newborns rubbed roughly and vigorously for extra-stimulation more times than recommended"

3. Newborns who were not put to breast within 1 hour of birth

4. Newborns who did not have their umbilical cords kept intact for at least 1 minute after birth

5. Newborns who received medical care non-consented by a parent or guardian

6. Unnecessary separation of mother and newborn defined as newborn not kept in skin to skin contact with mother immediately after birth

### Health system constraints

7. Newborns kept in a room at a temperature less than 25 C

8. Newborns delivered in a non-hygienic conditions, defined as health care staff who did not clean their hands clean their hands correctly as per the WHO's "five moments for hand hygiene"

9. Newborns born without inadequate preparation for birth and potential resuscitation defined as lack of equipment for immediate newborn care

### Socio-demographic, obstetric, and newborn characteristics

Women's ethnicity was categorized based on self-identified membership in groups according to the caste system [25] (i.e., relatively disadvantaged ethnic groups (*chhetri*, *janjati*, *madeshi*, *dalit other disadvantaged*) and relatively advantaged ethnic group (*brahmin*). Women' age was categorized as 18 years or less, 19–24 years, 25–29 years, 30–34 years, and 35 years or more. Women's literacy was categorized as those who were not literate and those were literate, based on self-report. Parity was categorized as women who had no previous births, at least one previous birth, and two or more previous births. Preterm birth was defined if the gestational age of birth was less than 37 completed weeks based on last menstrual period count. Sex of baby was defined as male or female.

### Constructing a mistreatment index

The distribution of the measures for mistreatment of newborns (Table 1) shows that the prevalence of some of the measures was more than 90% while for some it is less than 10%.

Given these conditions, if the mistreatment is defined based on the condition that if any of these 9 items have "Yes" then the mistreatment prevalence will be overestimated. Also, combining variables to binary ones would lose important information. To overcome this issue, we constructed a continuous score to represent the mistreatment index using principal component analysis (PCA) of the 9 items. PCA is a dimension reduction technique used for combining many variables into one. The usual practice is to weight the variable according to the first principal component i.e. the component which has the highest variance, and thus the highest discriminatory power. This single index is similar to a single factor in factor analysis setting. The continuous score is more flexible to analyse and to model. We consider the first principal

**Table 1. Prevalence of mistreatment of newborns after birth.**

| Indicator | Proportion (95% CI)* |
| --- | --- |
| Suctioning of breathing newborns | 21.4% (20.9–21.8) |
| Not treated with kindness and respect (roughly handled) | 25.0% (24.5–25.5) |
| No breast feeding within 1 hour | 73.1% (72.6–73.6) |
| Cord clamping within 1 minute | 73.2% (72.7–73.7) |
| Newborns not kept in skin to skin contact with mother immediately after birth | 96.7% (96.5–96.9) |
| Medical procedure without parental consent | 63.0% (62.5–63.5) |
| No hygiene maintained | 24.4% (23.9–24.8) |
| Temperature not adequately maintained | 44.9% (44.3–45.4) |
| No preparedness for birth and resuscitation | 57.8% (57.2–58.3) |

* For the quantiles of the actual variable.

component as the proxy for the mistreatment index as it explains more than thirty percent of the total variation in the mistreatment measure.

### Dealing with missingness

Among the potential covariates, we have missing values in the women's literacy and sex of the newborn. We graphically checked that the items distributions are not uniform between newborns with missing observations and without missing observations (S1 and S2 Figs). The same is also found in the distribution of PCA1 (S3 Fig). The t-test for the mean PCA1 was also found to be significant for both of the variables with p-values <0.001. This is an indication that missingness is not completely random. Therefore, we have imputed the missing values in the women's literacy and newborn sex utilizing the classification and regression tree (CART) method in the "mice" package in R.

### Regression analysis

The association between the mistreatment index and the covariates was assessed using graphical tools and the uni-variate linear and non-linear regression model. Potential covariates were maternal age, literacy, ethnicity, and parity, sex of the newborn, and term or preterm status of the newborn. These are selected based on a literature review and the availability of the data. The plot is suggestive that the selected covariates have an association with the mistreatment index. A non-linear association between maternal age and the mistreatment index was observed using Loess-smoothing on a scatter plot (S4 Fig) and two knots association was seen at 21 and 24 years using the broken-stick regression model (S2 Table). To avoid this ambiguity, linear regression was done for age category to assess the association with the mistreatment index.

For adjusting the effect of multiple covariates, we have fitted a multiple linear regression model including all the variables at the same time. After adjusting, the covariates still showed a significant association with the mistreatment index.

### Adjusting for between observer and between hospital heterogeneity

This study was conducted in 4 hospitals and 24 trained data collectors observed the labour and delivery. Therefore, there is a possibility that the observations between the observers and between the hospitals are heterogeneous and those within observers and within hospitals were correlated. In this situation, the standard error could be underestimated which produces a narrow confidence interval of the estimates. This leads to a false decision against the null hypothesis. To adjust for potential heterogeneity, we have employed a random intercept model. First, we considered the hospital as random variable and employed a mixed model. Secondly, we considered observers as random variable. We also had a third model, where we considered observers nested within the hospital as random variable. In the third model, there were three different levels (birth, observer, and hospital), and we were unable to get the convergence. The outcome from the first and second model is presented in Table 3 and the outcome of the first model has been reported in the main text.

All of the data analysis was conducted using R version 3.6.2 on the Linux operating system.

### Ethics approval and consent to participate

The main study was approved by ethical committee at Nepal Health Research Council NHRC (reference number 26–2017). Written consents were obtained from all the participants

included in the study prior to their participation. Written consent was obtained from a parent or legal guardian on behalf of the participants under the age of 18.

## Results

### Prevalence of mistreatment to newborns

Of the total 39,652 women admitted in labour for vaginal delivery in the hospitals during the study period, 34,125 of them consented and were enrolled in the study. Among their vaginally-born newborns, 31,804 breathed spontaneously at birth and were eligible for inclusion in the study (Fig 1).

Prevalence of mistreatment is shown in Table 1. Of the newborns who had spontaneous breathing (N = 31,804), 21.4% (95% CI, 20.9–21.8) received suctioning without medical indication. One fifth of the newborns, or 19.9% (95% CI, 19.4–20.4), were physically handled in a rough manner (lack of kindness and respect) and 19.4% (95% CI, 18.9–19.9) were born in non-hygienic conditions. Among these same newborns, 71.7% (95% CI, 71.2–72.3) had the cord clamped within 1 minute and 77.6% (95% CI, 77.1–78.1) were not breast fed within 1 hour of birth. Only 3.5% (95% CI, 3.2–3.8) were kept in skin to skin contact in the delivery room after birth. Among these newborns, 63.0% (62.5–63.5) of them received a medical intervention without consent from a parent or guardian. Further, 46.9% (95% CI, 46.3–47.6) of the newborns were not delivered in a room where the temperature was adequately maintained.

There was variation in the prevalence of indicators of mistreatment by women's age, ethnicity, education, parity, term or preterm infant status, and sex of the newborn (S5–S10 Figs).

Based on the first principal component analysis (Fig 2), a continuous mistreatment index between –3 to +3 was generated. The box plot of the mistreatment index shows variation with background characteristics of women and newborns. Maximum variation in the mistreatment index was observed among the infants born to women from advantaged ethnic groups and infants born to women who had previous two birth or more (Fig 3).

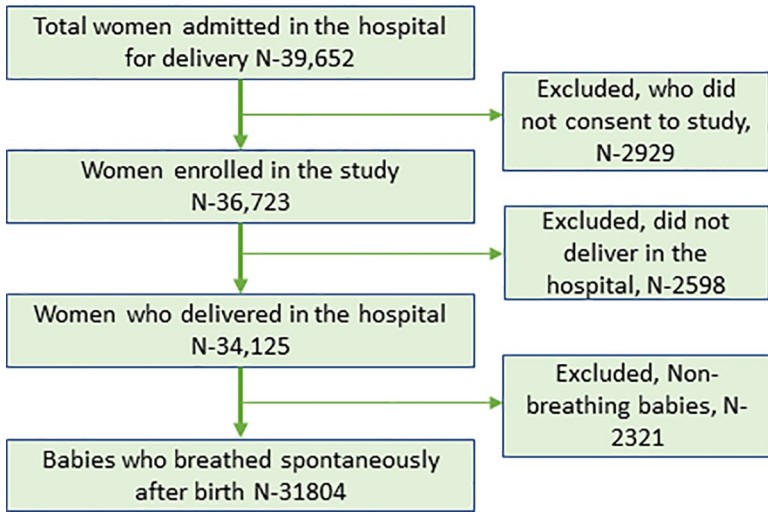

**Fig 1. Study flow diagram with number of participants.**

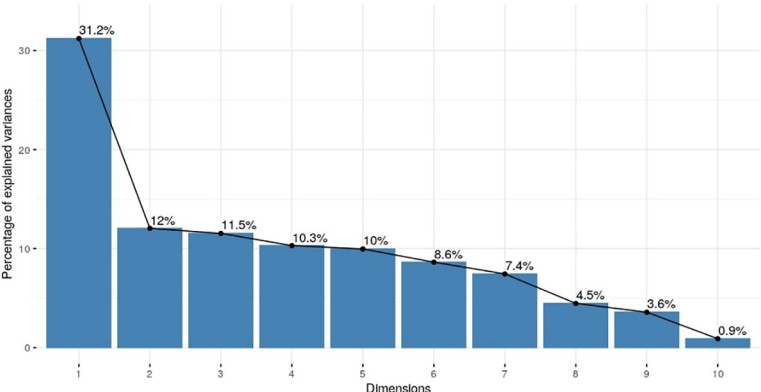

**Fig 2. Summary of principal component analysis.**

In multi-variate regression analysis, all of the covariates (maternal age, ethnicity, literacy, age, parity, sex of newborn, and term or preterm status) were found to be significantly associated with the mistreatment index even at a 5% level of significance.

Infants born to women aged 35 years or more were more likely to be mistreated ($\beta$, 0.466; p value, 0.000) in reference to infants born to women aged 18 years or less. Infants born to women of a relatively disadvantaged (*janjati*) ethnic group were more likely to be mistreated ($\beta$, 0.848; p value, 0.000) in reference to a relatively advantaged (*brahmin*) ethnic group. Infants born to women who had one previous birth ($\beta$, -0.176, p = 0.000) and infants born to women who had two or more previous births ($\beta$, -0.953; p value, 0.000) were less likely to be mistreated than those infants born to women who did not have a previous birth. Infants born to women who were illiterate ($\beta$, 0.425; p value, 0.000) were more likely to be mistreated than those born to literate women. Newborns who were born preterm ($\beta$, 0.280; p value, 0.000) were more likely to be mistreated than those born at term. Female newborns ($\beta$, 0.080; p value, 0.000) were more likely to be mistreated than male newborns (Table 2).

We adjusted the hospital heterogeneity through a random intercept model. The hospital-specific intercepts ($\beta$) were -2.87, 1.54, 0.95 and 0.38 at hospitals 1, 2, 3 and 4 respectively. After adjusting for hospital heterogeneity, we witnessed a change in the association between

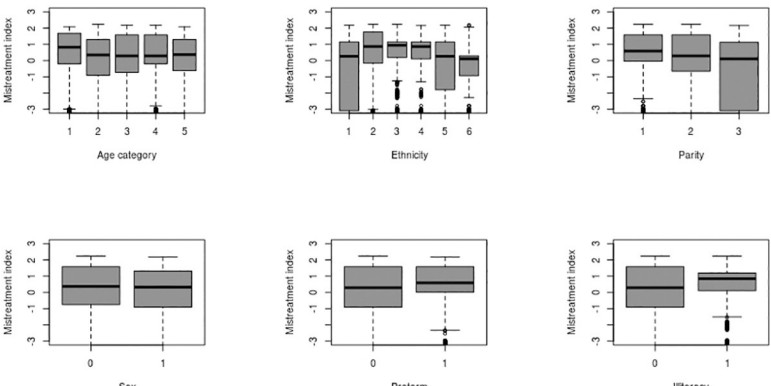

**Fig 3. The association between the mistreatment score and the covariates, assessed using graphical tools and a uni-variate regression model.** [¥]*Age category* 1 = 18 or less yrs, 2 = 19–24 yrs, 3 = 25–29 yrs, 4 = 30–34 yrs and 5 = 35 yrs and above. [§]*Ethnicity* 1 = advantaged ethnic group, 2 = dalit, 3 = janjati, 4 = Madeshi, 5 = other disadvantaged, 6 = Chhetri. [€]*Sex* 0 = male, 1 = email; [£]*Preterm* 0 = term, 1 = preterm; [±] *Illiteracy* 0 = literate, 1 = Illiterate.

**Table 2. Regression analysis of mistreatment among newborns (using mistreatment index).**

| | Unadjusted Estimate | | Adjusted by Other factors | |
|---|---|---|---|---|
| | **Beta** | **P-value** | **Beta** | **P-value** |
| **Age category** | | | | |
| *Intercept* | *0.184* | *0.000* | *-0.275$^{\$}$* | *0.000* |
| 18 year or less | Reference | | Reference | |
| 19–24 years | -0.233 | 0.000 | -0.093 | 0.021 |
| 25–29 years | -0.158 | 0.000 | 0.169 | 0.000 |
| 30–34 years | -0.114 | 0.022 | 0.310 | 0.000 |
| 35 years or above | -0.117 | 0.164 | 0.466 | 0.000 |
| **Ethnicity** | | | | |
| *Intercept* | *-0.366* | *0.000* | | |
| Relatively advantaged ethnic group (Brahmin) | Reference | | Reference | |
| Dalit | 0.559 | 0.000 | 0.520 | 0.000 |
| Janjati | 0.911 | 0.000 | 0.848 | 0.000 |
| Madhesi | 0.846 | 0.000 | 0.786 | 0.000 |
| Other disadvantaged | 0.203 | 0.000 | 0.190 | 0.000 |
| Chhetri | -0.028 | 0.596 | -0.024 | 0.649 |
| **Parity** | | | | |
| *Intercept* | *0.162* | *0.000* | | |
| 0 previous births | Reference | | Reference | |
| 1 previous births | -0.114 | 0.000 | -0.176 | 0.000 |
| 2–5 previous births | -0.779 | 0.000 | -0.953 | 0.000 |
| **Infant sex** | | | | |
| *Intercept* | *-0.0461* | *0.001* | | |
| Male | Reference | | Reference | |
| Female | 0.099 | 0.000 | 0.080 | 0.000 |
| **Preterm** | | | | |
| *Intercept* | *-0.031* | *0.003* | | |
| Term | Reference | | Reference | |
| Preterm | 0.309 | 0.000 | 0.280 | 0.000 |
| **Illiteracy** | | | | |
| *Intercept* | *-0.048* | *0.000* | | |
| Literate | Reference | | Reference | |
| Illiterate | 0.444 | 0.000 | 0.425 | 0.000 |

$^{\$}$Global intercept.

maternal and neonatal characteristics and the mistreatment index. There was no association between mistreatment with maternal literacy, parity, and preterm birth. Infants born to women aged 30–34 years (β, -0.041; p value, 0.01) and aged 35 years or more (β, -0.011; p value, 0.009) were less to be mistreated than women aged 18 years or less. Infants born to women of one relatively less advantaged (*janjati*) ethnic group (β, -0.042; p value, 0.002) were less likely to be mistreated than infants born to women of the relatively advantaged (*brahmin*) ethnic group. Infants born to women of another relatively less advantaged (*chhetri*) ethnic group (β, 0.077; p value, 0.000) were more likely to be mistreated than infants born to women of the relatively advantaged (*brahmin*) ethnic group. Female newborns (β, 0.016; p value, 0.015) were more likely to be mistreated than male newborns (Table 3).

**Table 3. Multi-level regression analysis on mistreatment among newborns using a mistreatment index (hospital and observer as random variable).**

|  | Adjusting hospital level heterogeneity (Model I) | | Adjusting observer level heterogeneity (Model II) | |
|---|---|---|---|---|
|  | Adjusted Beta | P-value | Adjusted Beta | P-value |
| **Age category** | | | | |
| *Intercept* | *-0.14258* | *0.88482* | *-0.15135* | *0.67142* |
| 18 or less | Reference | | Reference | |
| 19–24 years | -0.02350 | 0.09253 | -0.02493 | 0.0707 |
| 25–29 years | -0.02232 | 0.14536 | -0.01726 | 0.25367 |
| 30–34 years | -0.04181 | 0.01963 | -0.04524 | 0.01024 |
| 35 years or above | -0.06425 | 0.02945 | -0.05507 | 0.05935 |
| **Ethnicity** | | | | |
| Relatively advantaged ethnic group (Brahmin) | Reference | | Reference | |
| Dalit | -0.00085 | 0.93756 | 0.001116 | 0.91695 |
| Janjati | -0.04248 | 0.00214 | -0.04321 | 0.00155 |
| Madhesi | -0.02904 | 0.20658 | -0.0247 | 0.27581 |
| Other disadvantaged | -0.00464 | 0.66098 | -0.00758 | 0.46896 |
| Chhetri | 0.07792 | 0.00002 | 0.077344 | <0.001 |
| **Parity** | | | | |
| 0 previous births | Reference | | Reference | |
| 1 previous births | 0.00292 | 0.71485 | 0.000314 | 0.96974 |
| 2–5 previous births | 0.01014 | 0.33817 | 0.061263 | <0.001 |
| **Infant Sex** | | | | |
| Male | Reference | | Reference | |
| Female | 0.01613 | 0.01595 | 0.010925 | 0.01768 |
| **Preterm** | | | | |
| Term | Reference | | Reference | |
| Preterm | 0.00974 | 0.37864 | 0.010124 | 0.35551 |
| **Illiteracy** | | | | |
| Literate | Reference | | Reference | |
| Illiterate | 0.00968 | 0.40681 | -0.00433 | 0.70328 |

## Discussion

The study reports on the prevalence of mistreatment after birth of healthy vaginally-born newborns in health facilities. Unnecessary suctioning was seen in one fifth of spontaneously breathing infants. One fourth of the newborns were not treated with respect and dignity (roughly handled) and same proportion were born in places where basic hygiene was not maintained at birth. Two-thirds of the newborns were provided a medical intervention without consent from a parents or guardian. Almost half of the infants were born in a room with temperature not adequately maintained. Mistreatment of newborns was associated with being born to women aged less than 18 years, being born to women from disadvantaged ethnic groups, and if the newborn was female.

The prevalence of mistreatment of newborns and the variance of mistreatment by socio-demographic characteristics suggests that changes are required at the health system level, among individual health workers, and around social norms. Clinical practices such as maintaining hygiene and preparing the necessary equipment and environment for birth and potential complications will require health system support on availability of equipment and commodities [26], while individual health worker poor performance such as not encouraging the newborn to be kept in skin to skin contact immediately after birth will require capacity

building on clinical standards, and reducing discrimination towards newborns based on maternal and newborn characteristics will require training on medical ethics and implicit bias. Additional changes to ensure respect for health workers may in turn increase their ability to provide respectful care for their patients.

The use of unnecessary interventions such as suctioning to a spontaneously breathing newborns has been observed in a previous study on neonatal resuscitation [26]. The prevalent use of suctioning might be due to the pre-existing clinical practice to remove secretions, but is no longer recommended in spontaneously breathing infants [27]. Newborns in this study who were spontaneously breathing also received unnecessary and vigorous stimulation, which is not evidence-based [28, 29].

Despite the medical ethical guidance for all health workers to take consent from parents or guardian to treat any infant, there is a poor adherence to the guidance [28]. A cross-sectional study conducted in 4 health facilities in Addis Ababa to measure disrespect and abuse during childbirth showed the rights to information, informed consent, and choice/preference were not protected in nearly 94.8% of women. Almost half of the women were not asked for their consent or permission prior to any procedure for themselves or infants [30]. Parents or other legal guardians have the right to consent for care for their newborns. Health workers will require training to improve their skills to inform women and consent for care being provided to newborn infants.

This study found more mistreatment of infants born to younger women. Adolescent pregnancy in Nepal has been stigmatized in Nepal [31] and may be contributing to health worker attitudes toward young women during childbirth. More mistreatment of young pregnant women might also be due to the fact that these women are less educated and are less aware of their rights during childbirth [32]. This correlates findings from another study in Nepal which showed that mistreatment of women during childbirth was more reported among younger age women.

This study found that the risk of mistreatment for newborns increased when born to women of relatively disadvantaged ethnic groups. In Nepal, caste and ethnicity remains the center-piece of the social hierarchy [25]. The study found that newborns born to women in the *janjati* ethnic group are at a relative risk of mistreatment in reference to those born to women in a relatively advantaged ethnic group. The study did not find any association with other relatively disadvantaged groups (*dalit*, *madeshi* and other disadvantaged). This might be due to health worker attitudes towards women from *janati* ethnic group or might be because health workers were from different ethnic groups from their patients and were not trained to provide culturally appropriate care. Further exploration will be required to understand the provider and client dynamics based on ethnicity. In this study, we observed mistreatment of newborns based on their sex, demonstrating that gender remains an important barrier for receiving respectful care in health facilities, even from the time of infancy. In Nepal, there is a strong male preference, which has led to serious discriminatory behaviour towards female infants [33] even at the health institution settings, as evident in this study.

## Methodological considerations

This study has several strengths including that observations were done by trained researchers in multiple hospitals in Nepal in a consistent way and with a large sample size. Using observational methods eliminated the potential for recall bias, and use of both observations and a semi-structured interview strengthened confidence in the findings.

However, there are several limitations. First, observations were only done for vaginal births, not caesarean section births. Exclusion of births by caesarean section introduces the possibility

of selection bias as it excluded infants born to mothers who had delivery using general anaesthesia or an intrapartum related complication. Second, there might have been observer reported bias, especially when the caseload for births were high, possibly resulting in fewer observations of mistreatments. Periodic quality assessment was done to reduce the bias. Third, there could have been some subjectivity in the reporting from the study nurses on mistreatment observations, especially for indicators, such as "newborn handled roughly" and "newborn received non-consented medical intervention." It may have been helpful to also collect data anonymously through caregiver self-report following discharge in a neutral environment; however, this was not feasible as part of this study. Fourth, we did not assess some health system constraints, such as number of women and newborns taken care of by each nurse per day or other facility assessments. Fifth, due to the high level of heterogeneity in process of care for newborns by hospitals, the association between mistreatment with maternal socio-demographic and neonatal characteristics changed after adjusting hospital level heterogeneity. Finally, there were large numbers of missing values for women's literacy and sex of newborn and the values missing were not distributed uniformly and randomly to exclude the missing value from analysis, so imputation was done.

## Conclusion

The prevalence of mistreatment of healthy, spontaneously-breathing newborns is high in public hospitals of Nepal. The prevalence of mistreatment varies by hospital, women's ethnicity, and women's age, as well as sex of the newborn. Further studies will be required to investigate the variance of mistreatment by health facilities and implementation studies will be required to identify the most effective interventions to reduce inequity and mistreatment.

## Supporting information

**S1 Checklist. STROBE 2007 (v4) statement—checklist of items that should be included in reports of *cohort studies*.**
(DOCX)

**S1 Data.**
(CSV)

**S1 Fig.**
(TIF)

**S2 Fig.**
(TIF)

**S3 Fig.**
(TIF)

**S4 Fig.**
(TIF)

**S5 Fig.**
(TIF)

**S6 Fig.**
(TIF)

**S7 Fig.**
(TIF)

**S8 Fig.**
(TIF)

**S9 Fig.**
(TIF)

**S10 Fig.**
(TIF)

**S1 Table. Definition of mistreatment among newborns during childbirth.**
(DOCX)

**S2 Table. Multi-level regression analysis on mistreatment among newborns (hospital and observer as random variable).**
(DOCX)

**S1 Study.**
(PDF)

## Acknowledgments

We would like to thank Omkar Basnet, database manager of Golden Community, for data curation.

## Author Contributions

**Conceptualization:** Ashish K. C.

**Data curation:** Avinash K. Sunny.

**Formal analysis:** Md Moinuddin, Avinash K. Sunny.

**Funding acquisition:** Emma Sacks, Mats Målqvist.

**Investigation:** Ashish K. C., Pratiksha Bhattarai.

**Methodology:** Ashish K. C., Mary Kinney, Emma Sacks, Srijana Sharma, Mats Målqvist.

**Project administration:** Rejina Gurung, Pratiksha Bhattarai, Srijana Sharma.

**Resources:** Mats Målqvist.

**Software:** Ashish K. C.

**Supervision:** Rejina Gurung, Pratiksha Bhattarai, Srijana Sharma.

**Validation:** Ashish K. C.

**Writing – original draft:** Ashish K. C.

**Writing – review & editing:** Ashish K. C., Mary Kinney, Emma Sacks, Rejina Gurung, Mats Målqvist.

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
