## [Decision Letter · Decision Letter 0]

1 Oct 2020

PONE-D-20-21414

Mistreatment among newborns after childbirth in health facilities in Nepal: results from a prospective cohort observational study

PLOS ONE

Dear Dr. KC,

Thank you for submitting your manuscript to PLOS ONE. After careful consideration, we feel that it has merit but does not fully meet PLOS ONE’s publication criteria as it currently stands. Therefore, we invite you to submit a revised version of the manuscript that addresses the points raised during the review process.

On of the reviewers raised question about ethics.

Please first of all  provide the accordance and the protocol of the ethics commission to clarify  this crucial point.

We look forward to receiving your revised manuscript.

Kind regards,

David Desseauve, MD, MPH, PhD

Academic Editor

PLOS ONE

Journal Requirements:

a) Did participants provide their written or verbal informed consent to participate in this study?

Reviewers' comments:

Reviewer's Responses to Questions

**Comments to the Author**

1. Is the manuscript technically sound, and do the data support the conclusions?

Reviewer #1: Yes

Reviewer #2: Yes

2. Has the statistical analysis been performed appropriately and rigorously? 

Reviewer #1: I Don't Know

Reviewer #2: Yes

3. Have the authors made all data underlying the findings in their manuscript fully available?

Reviewer #1: No

Reviewer #2: No

4. Is the manuscript presented in an intelligible fashion and written in standard English?

Reviewer #1: No

Reviewer #2: Yes

5. Review Comments to the Author

Reviewer #1: Thank you for allowing me to review the manuscript entitled “Mistreatment among newborns after childbirth in health facilities in Nepal: results from a prospective cohort observational study”.

In this paper, Ashish et al. report a large, prospective, multicentric cohort conducted in 4 public hospitals in Nepal and including 31’804 newborns. This observational study nested in a larger study on neonatal resuscitation aimed to assess frequency and risk factors of mistreatments at childbirth among vaginally-born spontaneous breathing newborns. Results show a high rate of inadequate care, notably, that nearly two thirds of newborns received medical interventions without parental consent, 25% were handled roughly, and 21% were suctioned without need. Main risk factors associated to mistreatment were ethnicity, young maternal age, and newborn female sex. Authors also propose a set of indicators based on the WHO standards to assess neonatal mistreatment.

This study is one of the first investigating the question of “mistreatment” of newborns at birth. It is indeed an interesting and relevant topic, and it deserves not only to be published, but also to be reproduced in other care settings, including high income countries.

I have the following questions and comments to address to the authors:

1. General: the article is well and logically written, but it is too long and will gain in being more concise and straight, in particular: abstract (2 pages), author summary, introduction, discussion (4 pages)

2. Abstract: information on cord clamping, breast-feeding during the first hour and skin to skin contact would be important to report.

3. Methods: more information about the four hospitals would be interesting (number of yearly deliveries, rates of C-section, number of midwifes, nurses and physicians)

4. Statistics: complex and not easy to follow for a non-statistician. Anyway, it seems it could be interesting to add a composite dichotomic variable of “any” or “no” mistreatment to know which proportion of babies are concerned, even (or especially!) if very high.

5. Discussion, page 20, lines 20-21: “Newborns who were spontaneously breathing also received unnecessary and vigorous stimulation, which is not evidence-based”. In the methods, it is set that non spontaneously breathing was a criteria exclusion.

Minor:

6. Abstract, page 3, line2: “women age 35 or more were more likely to be mistreated” � “Infants born to women (?!)were more likely to be mistreated”

7. Abstract, results, pages 2-3 and author summary, pages 5-6: presentations of factors should be in the same way, either increasing or decreasing the risk of mistreatment, but mixing the two is somewhat confusing for the reader.

8. Method, page 14, lines 1-2: “The single index similar to a single factor analysis setting”: missing verb?

9. Method, page 15, line 15-19: “In the third model, there were three different levels (birth, observer and hospital), so the model became complicatenand we were unable to get the convergence and The outcome from the first and second 18 model is presented in table 3 and the outcome of the first model has been reported in the 19 main text: missing punctuation?

Reviewer #2: Reviewer N° 4

Manuscript entitled “Mistreatment among newborns after childbirth in health facilities in Nepal: results from a prospective cohort observational study” is a well written interesting manuscript focusing on mistreatment of mothers and newborn infants just after birth.

The results of this study are a side product of a prospective cohort observational study of around 80’000 deliveries conducted over a period of 18 months in 12 hospitals in Nepal.

The relationship of the here presented data to the initial study is not quite clear and leads to some major questions:

1. Consent: It is not quite clear what was included in the consent obtained by the mothers. Some fundamental questions might arise and should be cleared before publishing this paper:

a) One of the major results is the lack of consent for treatment issues in spontaneously breathing infants. It seems difficult to understand that mothers who consented for this study should be asked for additional consent for each treatment approach.

b) Was stated in the consent form that the well-trained study nurse would not intervene in the treatment of the newborn infant, although this treatment might be misconducted and not performed according to the international standards?

2. Hospital selection? The data presented in the abstract demonstrate that the influence of hospitals might have an important impact on the results, as adjustment for hospital heterogeneity inverse the results from more to less mistreatment in women over 35 years.

Abstract:

Methods and Findings:

In the second paragraph is stated: In multi-variate regression analysis, women age 35 years or more were more likely to be mistreated (β, 0.466; p value, 0.000) in reference to 18 years or less.

In the next paragraph is stated: After adjusting for hospital heterogeneity, women aged 30-34 years (β, -0.041; p value, 0.01) and aged 35 years or more (β, -0.064; p value, 0.029) were less mistreated in reference to women aged 18 years or less.

I could not find any explanation, why this study was performed only in four out of the twelve hospitals and how they have been selected.

3. Cesarean section: In Figure 1 the flow chart is given for the 39’652 women admitted to one of the four hospitals, leading to the 31’804 deliveries, which were observed. The abstract indicates as one major limitations that newborns delivered by caesarean section have not been included. The number of children born by CS should be given and substracted from the 31’804 deliveries in order to get the final denominator.

4. Evolution over time: The goal of the original study was to measure the impact of training regarding quality improvement in mother and newborn care. No information in the here presented study is given regarding the timely evolution over the 18 months study period. There is a potential study bias, which should unveiled.

Minor points:

Methods:

Design: April 2017 – October 2018 = 19 months (exact data of study period should be stated).

Final remark:

I do not comment on the questions and remarks of the other reviewers, which I fully agree with.

I would conclude that the study unveils important lack of newborn care in four Nepalese Hospitals which might point to similar problems in many other maternity wards worldwide.

There are systematic problems which should be taught differently:

o late cord clamping and breast feeding within the first hour seems not to be part of the teaching standards as it was not performed in > 73% of situations.

o Even more the fact that the well-being newborn is separated from the mother in > 95% of all vaginal deliveries.

o The question of informed consent needs to be approached differently.

o Temperature control in the delivery room is in 50% of the hospitals not given.

o Preparedness and hygiene is important to increase

Therefore, the information of this study is important. However, the abovementioned major questions should be answered before publication.

6. PLOS authors have the option to publish the peer review history of their article (what does this mean?). If published, this will include your full peer review and any attached files.

Reviewer #1: No

Reviewer #2: No

---

## [Author Response · Author response to Decision Letter 0]

30 Oct 2020

-Response to editor’s comment

Comment 1# - Please first of all provide the accordance and the protocol of the ethics commission to clarify this crucial point.If applicable, we recommend that you deposit your laboratory protocols in protocols.io to enhance the reproducibility of your results. Protocols.io assigns your protocol its own identifier (DOI) so that it can be cited independently in the future. For instructions see: http://journals.plos.org/plosone/s/submission-guidelines#loc-laboratory-protocols

Response-We have attached the ethics approval provided by the ethics review committee of Nepal Health Research Council. We have also attached the study protocol published in BMJ protocol. Please note that this study is a nested study within a larger study that evaluated health worker’s performance on neonatal care. We use the observational data on health worker’s performance as per the protocol to assess various newborn care practices that constitute mistreatment. We have also deposited the data and data analysis protocol.

Response to first reviewer’s comments

Comment 2# General: the article is well and logically written, but it is too long and will gain in being more concise and straight, in particular: abstract (2 pages), author summary, introduction, discussion (4 pages)

Response- We have now shortened the abstract and body of the article.

Comment 3# Abstract: information on cord clamping, breast-feeding during the first hour and skin to skin contact would be important to report.

Response- We have added the information on cord clamping, breast-feeding during the first hour and skin to skin contact to the abstract.

Comment 4# Methods: more information about the four hospitals would be interesting (number of yearly deliveries, rates of C-section, number of midwifes, nurses and physicians)

Response- We have now added the more information in page 9, lines 17-22 and page 10, lines 1-2.

Comment 5# Statistics: complex and not easy to follow for a non-statistician. Anyway, it seems it could be interesting to add a composite dichotomic variable of “any” or “no” mistreatment to know which proportion of babies are concerned, even (or especially!) if very high.

Response- Thank you for noting this. We have simplified our explanation of the methods and have added a simple summary at the start of the results section. However, we have retained some explanations of the statistics as this is an academic journal and these details are required. We created a scoring index of mistreatment using the principal component analysis of the childbirth care related-indicators, and conducted linear regression based on the mistreatment index. Therefore, it will not be possible to dichotomize the mistreatment index.

Comment 6# Discussion, page 20, lines 20-21: “Newborns who were spontaneously breathing also received unnecessary and vigorous stimulation, which is not evidence-based”. In the methods, it is set that non spontaneously breathing was a criteria exclusion.

Response- Correct, newborns who were not spontaneously breathing were excluded as they require additional (e.g. not routine) care. We included only newborns who spontaneously breathed at birth. These newborns do not require any resuscitation i.e. stimulation. Therefore, providing unnecessary and vigorous stimulation to these spontaneously breathing newborns is not evidence-based. 

Comment 7# Abstract, page 3, line2: “women age 35 or more were more likely to be mistreated” � “Infants born to women (?!)were more likely to be mistreated”

Response- Thank you. We have now revised it as “infants born to” women etc.

Comment 8# Abstract, results, pages 2-3 and author summary, pages 5-6: presentations of factors should be in the same way, either increasing or decreasing the risk of mistreatment, but mixing the two is somewhat confusing for the reader.

Response- We have now ensured that the presentations of factors in abstract and summary are presented in the same direction.

Comment 9# Method, page 14, lines 1-2: “The single index similar to a single factor analysis setting”: missing verb?

Response- We have now added the verb.

Comment 10# Method, page 15, line 15-19: “In the third model, there were three different levels (birth, observer and hospital), so the model became complicated and we were unable to get the convergence and The outcome from the first and second 18 model is presented in table 3 and the outcome of the first model has been reported in the 19 main text: missing punctuation?

Response- We have added a punctuation.

Response to second reviewer’s comments

Comment 11# The relationship of the here presented data to the initial study is not quite clear and leads to some major questions. Consent: It is not quite clear what was included in the consent obtained by the mothers. Some fundamental questions might arise and should be cleared before publishing this paper.

Response- We have clarified the questions on consent in the below queries. 

Comment 12# One of the major results is the lack of consent for treatment issues in spontaneously breathing infants. It seems difficult to understand that mothers who consented for this study should be asked for additional consent for each treatment approach.

Response-Women consented to the larger study which was to be observed for research purposes by an independent observer, but this was different than the providers seeking consent for treatment or care (Of self or newborn).

Comment 13# Was stated in the consent form that the well-trained study nurse would not intervene in the treatment of the newborn infant, although this treatment might be misconducted and not performed according to the international standards?

Response- We recognize the challenge for clinicians in the role of non-participating observers. In this case, the trained study nurses were trained on clinical observation and recording and not on clinical skills to intervene in care during childbirth and immediately after birth. In condition of life-threatening emergency of women and newborn infant, the well-trained study nurses were trained to inform the clinicians.

Comment 14# Hospital selection? The data presented in the abstract demonstrate that the influence of hospitals might have an important impact on the results, as adjustment for hospital heterogeneity inverse the results from more to less mistreatment in women over 35 years.

Response- For the larger study, 23 hospitals were selected. It was a clustered study and the highest volume hospital in each study served as the site for observations. For this study, only those 4 hospitals where clinical observations were done were selected. 

Since the intervention time in these hospitals differed from each other, there was variation in the quality of care. This is reflective in the results, as the results changed when hospitals were adjusted. The change in the result after adjusting the hospital heterogeneity indicates that the coverage of respectful care is not homogenous by hospital.

Comment 15# Abstract: Methods and Findings: In the second paragraph is stated: In multi-variate regression analysis, women age 35 years or more were more likely to be mistreated (β, 0.466; p value, 0.000) in reference to 18 years or less. In the next paragraph is stated: After adjusting for hospital heterogeneity, women aged 30-34 years (β, -0.041; p value, 0.01) and aged 35 years or more (β, -0.064; p value, 0.029) were less mistreated in reference to women aged 18 years or less. I could not find any explanation, why this study was performed only in four out of the twelve hospitals and how they have been selected.

Response- In abstract, to avoid the confusion, we presented only the results after adjusting with hospital heterogeneity. We removed the paragraph “In multi-variate regression analysis, women age 35 years or more were more likely to be mistreated (β, 0.466; p value, 0.000) in reference to 18 years or less.” We have explained in the response to comment 13 on why 4 out of 12 hospitals were selected. 

Comment 16# Cesarean section: In Figure 1 the flow chart is given for the 39’652 women admitted to one of the four hospitals, leading to the 31’804 deliveries, which were observed. The abstract indicates as one major limitations that newborns delivered by caesarean section have not been included. The number of children born by CS should be given and substracted from the 31’804 deliveries in order to get the final denominator.

Response- The figure 1, 39,652 women are those women who were admitted in the labour room for vaginal birth for observation. This does not include women who were taken to operation theatre directly for caesarean section. Those women who were admitted to labour room and take to operation theatre for emergency caesarean section were excluded in the analysis. We have corrected the typo as “excluded, did not deliver in delivery room”

Comment 17# Evolution over time: The goal of the original study was to measure the impact of training regarding quality improvement in mother and newborn care. No information in the here presented study is given regarding the timely evolution over the 18-month study period. There is a potential study bias, which should unveiled.

Response- We agree the study is a nested study and the aim of the larger study was to impact quality improvement of a neonatal resuscitation intervention on health worker’s performance and birth outcome. the larger study was targeting non-spontaneous breathing infants and not expected to have an impact on routine care. However, we have noted the possible impact of change over time in the limitations. months has been mentioned in page 9, lines 5-7.

Comment 18# Methods: Design: April 17, 2017 – October 2018 = 19 months (exact data of study period should be stated).

Response- It is an 18 month study with time point 14 April 2017 and 17 October 2018. We have now added it.

Comment #19:

I do not comment on the questions and remarks of the other reviewers, which I fully agree with.

I would conclude that the study unveils important lack of newborn care in four Nepalese Hospitals which might point to similar problems in many other maternity wards worldwide.

There are systematic problems which should be taught differently:

o late cord clamping and breast feeding within the first hour seems not to be part of the teaching standards as it was not performed in > 73% of situations.

o Even more the fact that the well-being newborn is separated from the mother in > 95% of all vaginal deliveries.

o The question of informed consent needs to be approached differently.

o Temperature control in the delivery room is in 50% of the hospitals not given.

o Preparedness and hygiene is important to increase

Therefore, the information of this study is important. However, the abovementioned major questions should be answered before publication.

Response- Thank you. We have endevoured to answer the above questions from reviewers, as we also believe this study has important findings for the Nepalese health system and newborns.

Comment #20 PLOS authors have the option to publish the peer review history of their article (what does this mean?). If published, this will include your full peer review and any attached files.

Response- We are fine with the publication of the peer review history for this manuscript.

---

## [Decision Letter · Decision Letter 1]

1 Dec 2020

PONE-D-20-21414R1

Mistreatment of newborns after childbirth in health facilities in Nepal: results from a prospective cohort observational study

PLOS ONE

Dear Dr. KC,

Thank you for submitting your manuscript to PLOS ONE. After careful consideration, we feel that it has merit but does not fully meet PLOS ONE’s publication criteria as it currently stands. Therefore, we invite you to submit a revised version of the manuscript that addresses the points raised during the review process.

ACADEMIC EDITOR:

Thank you for this revised version of the manuscript which improve the quality and the message of thsi manuscript. Nevertheless, some majors reviewers comments have to be answered clearly. 

In this form and without clear explanations about ethical consideartion and methodology this article couldn't be accepted. We invite you to provide us a newe version of this manuscript answering clearly each point arised by the reports reviewers  below.

We look forward to receiving your revised manuscript.

Kind regards,

David Desseauve, MD, MPH, PhD

Academic Editor

PLOS ONE

Reviewers' comments:

Reviewer's Responses to Questions

**Comments to the Author**

1. If the authors have adequately addressed your comments raised in a previous round of review and you feel that this manuscript is now acceptable for publication, you may indicate that here to bypass the “Comments to the Author” section, enter your conflict of interest statement in the “Confidential to Editor” section, and submit your "Accept" recommendation.

Reviewer #1: All comments have been addressed

Reviewer #2: (No Response)

2. Is the manuscript technically sound, and do the data support the conclusions?

Reviewer #1: Yes

Reviewer #2: Partly

3. Has the statistical analysis been performed appropriately and rigorously? 

Reviewer #1: I Don't Know

Reviewer #2: Yes

4. Have the authors made all data underlying the findings in their manuscript fully available?

Reviewer #1: Yes

Reviewer #2: No

5. Is the manuscript presented in an intelligible fashion and written in standard English?

Reviewer #1: Yes

Reviewer #2: Yes

6. Review Comments to the Author

Reviewer #1: In this revised version, authors adressed most of the comments of the reviewer. The current manuscript deserves to be published, regarding notably the new and important insights of the study on neonatal mistreatment at birth.

I have only 2 additional comments :

1- more emphasis could be brought on the mots frequent mistreatment , id est lack of skin to skin contact immediatly after birth that concerns almost all of the infants and could be easy to improve; what was the reason/barrier for this delay?

2- a possible impact of observation on care givers behaviour could be mentioned as a study limitation and possible bias

Reviewer #2: Thanks for giving me the opportunity to review the revised manuscript.

A general problem remains: This study seems more to describe a health care system in which evidence based neonatal basic care is almost absent, than a voluntarily applied mistreatment of newborn infants. The title might be reconsidered.

This study is important as it shows an absence of knowledge regarding neonatal care and an important need for improvement. However, to declare it as mistreatment as suggested in the second paragraph of the Background chapter, might be questionable. The main information of this study is comprised in Table 1 and the description in the first paragraph of the discussion (page 18, line 3 - 9).

However, although highly advanced statistical methods have been applied, the interpretation of the results are not always clear or are laking. Therefore several questions remain:

1. Sex of infants : In lines 5ff of page 17 is stated : « There was no association between mistreatment with woman’s literacy, parity, and newborn sex ». In line 13 is stated: « Female newborns were more likely to be mistreated than male newborns. » What is now true?

2. After adjustment, there is a change of the risk being mistreated for infants of mothers > 35 years, initially higher, after adjustment lower risk. Explanation is still missing for this risk change.

3. Page 15 lines 14/15 : Statement about mistreatment of women, however, the study is about their newborns. This should be clarified.

4. There is a statement that newborns of mothers of less advantaged ethnic groups are at higher risk for mistreatment. However, there is no explanation why newborns of mothers of the ‘janjati’ ethnic group, although less advantaged are at lower risk for mistreatment. Hypothèses?

Furthermore, there remain some more questions which were not clearly answered.

1. What exactly is the content of the parental information? This is important, as one of the major finding of this study is the non-consented treatment of the newborn infants. Although it is stated in the revised manuscript that consent was given to observation and not for treatment, this difference might be difficult to be understood by mothers and also by health care workers.

One conclusion of this point is to be made: information of mothers about neonatal care should be improved, but for this being possible, health care workers should be formed about basic skills and trained in their application.

2. Caesarean section and clear definition of the denominator: it is still not clearly written that there are only newborns after vaginal birth included.

Page 14 line 9ff :

It is stated: “Of the total 39,652 women admitted for delivery in the hospitals during the study period, 34,125 of them consented and were enrolled in the study. Among their newborns, 31,804 breathed spontaneously at birth and were eligible for inclusion in the study (figure 1).”

However: there was a caesarean section rate of 18 – 27%. Numbers have to be explained.

3. The statement on page 19, lines 13ff has to be changed : « The use of unnecessary interventions such as suctioning to a spontaneously breathing newborns has been observed in previous study on neonatal resuscitation[26]. Reducing this intervention would also allow delayed cord clamping, early initiation of breast-feeding, and immediate skin to skin contact. » The suctioning is not responsible for dealyed breat-feeding and immediate skin to skin contact.

Taken together :

The results of the highly performed statistics are not sufficiently explained. The important findings of this study are the fact that basic neonatal care is mainly absent in the four major maternity wards of Nepal and that there is a tremendous need for training of neonatal care algorithms and skills.

7. PLOS authors have the option to publish the peer review history of their article (what does this mean?). If published, this will include your full peer review and any attached files.

Reviewer #1: No

Reviewer #2: No

---

## [Author Response · Author response to Decision Letter 1]

28 Dec 2020

28 December 2020

Response to first reviewer’s comment

Comment 1#. more emphasis could be brought on the most frequent mistreatment, esp lack of skin to skin contact immediately after birth that concerns almost all of the infants and could be easy to improve; what was the reason/barrier for this delay?

Response. The lack of skin to skin contact immediately after birth might be due to health workers lack of knowledge on the importance of skin to skin contact immediately after birth or high caseload of birth. Further, qualitative assessment is needed to identify the factors for non-adherence to standards. To improve skin to skin contact immediately after birth there is a need to build capacity on clinical standards. We have mentioned this in page 19, lines 8-9.

Comment 2#. A possible impact of observation on care givers behaviour could be mentioned as a study limitation and possible bias

Response. We have provided the possible bias of observers reporting in limitation section.

Response to second reviewer’s comment

Comment 3#. A general problem remains: This study seems more to describe a health care system in which evidence based neonatal basic care is almost absent, than a voluntarily applied mistreatment of newborn infants. The title might be reconsidered.

Response. We agree that the study describes the non-adherence to standard care for newborn at the time of birth within a health system context. We define mistreatment as the care which deviates from the standard of care.

Comment 4#. This study is important as it shows an absence of knowledge regarding neonatal care and an important need for improvement. However, to declare it as mistreatment as suggested in the second paragraph of the Background chapter, might be questionable. The main information of this study is comprised in Table 1 and the description in the first paragraph of the discussion (page 18, line 3 - 9).

Response. This study provides new evidence on the deviation of clinical practice for newborn during childbirth. We have defined mistreatment in the method section, page 10, lines 9-10.

Comment 5#. Interpretation of the results are not always clear or are lacking. Therefore several questions remain: # Sex of infants: In lines 5ff of page 17 is stated : « There was no association between mistreatment with woman’s literacy, parity, and newborn sex ». In line 13 is stated: « Female newborns were more likely to be mistreated than male newborns. » What is now true?

Response. There was a typo. In page 17, line 6 it should be stated “There was no association between mistreatment with woman’s literacy, parity and premature birth” rather than ”…. newborn sex”. We have now revised it.

Comment 6#. Interpretation of the results are not always clear or are lacking. After adjustment, there is a change of the risk being mistreated for infants of mothers > 35 years, initially higher, after adjustment lower risk. Explanation is still missing for this risk change.

Response. we have now added the explanation on the change in the results on association of between mistreatment with socio-demographic and neonatal characteristics in limitation section in page 21, lines 20-23 as “Fifth, due to the high level of heterogeneity in process of care for newborns by hospitals, the association between mistreatment with socio-demographic and neonatal characteristics (age, literacy, parity and prematurity) changed after adjusting hospital level heterogeneity.”

Comment 7#. Interpretation of the results are not always clear or are lacking. Page 15 lines 14/15 : Statement about mistreatment of women, however, the study is about their newborns. This should be clarified.

Response. Thank you for the observation. We have now improved it as “Infants born to …..”

Comment 8#. Interpretation of the results are not always clear or are lacking. There is a statement that newborns of mothers of less advantaged ethnic groups are at higher risk for mistreatment. However, there is no explanation why newborns of mothers of the ‘janjati’ ethnic group, although less advantaged are at lower risk for mistreatment. Hypothèses?

Response. We have now explained it in page 20, lines 13-20. “The study found that of the ethnic group, only newborn born to women janjati ethnic group are at a relative risk of mistreatment in reference to relatively advantaged ethnic group. The study did not find any association with other relatively disadvantaged group (dalit, madeshi and other disadvantaged). This might be due to health worker’s attitude towards women from janati ethnic group or might be health workers were from other ethnic group and could not provide care culturally appropriate care. Further exploration will be required to understand the provider and client dynamics based on ethnicity.” 

Comment 9#. What exactly is the content of the parental information? This is important, as one of the major finding of this study is the non-consented treatment of the newborn infants. Although it is stated in the revised manuscript that consent was given to observation and not for treatment, this difference might be difficult to be understood by mothers and also by health care workers. One conclusion of this point is to be made: information of mothers about neonatal care should be improved, but for this being possible, health care workers should be formed about basic skills and trained in their application.

Response. We notice the confusion on the consent of parental information. There are two kind of consent for parental information. For, the research ethics purpose, consent was taken by our independent observers from women to observe them during clinical care process. For the clinical care purpose, observation was done whether health workers informed women and consented for any additional care provided to newborns. We agree to improve parental information, health workers skills needs to be developed. We have added a sentence in page 20, lines 4-6 as “Health workers will require training to improve their skills to inform women and consent for care being provided to newborn infants.”

Comment 10#. Caesarean section and clear definition of the denominator: it is still not clearly written that there are only newborns after vaginal birth included. Page 14 line 9ff : It is stated: “Of the total 39,652 women admitted for delivery in the hospitals during the study period, 34,125 of them consented and were enrolled in the study. Among their newborns, 31,804 breathed spontaneously at birth and were eligible for inclusion in the study (figure 1).” However: there was a caesarean section rate of 18 – 27%. Numbers have to be explained.

Response. Thank you. As explained in the inclusion criteria in method section, page 9 lines 1-2. The study participants were “all vaginally born newborns who were spontaneously breathing at birth…” In the setting in page 8, lines 18-19 we provide the context that in these four hospitals the caesarean section rate was 18-27%. We have clarified in the result section page 14, line 9-11 as “ Of the total 39,652 women admitted in labour room for vaginal delivery …../ Among their vaginally born newborns….”

Comment 11#. The statement on page 19, lines 13ff has to be changed : « The use of unnecessary interventions such as suctioning to a spontaneously breathing newborns has been observed in previous study on neonatal resuscitation[26]. Reducing this intervention would also allow delayed cord clamping, early initiation of breast-feeding, and immediate skin to skin contact. » The suctioning is not responsible for dealyed breast-feeding and immediate skin to skin contact.

Response. We have now removed the sentence.

Comment 12#. Taken together: The results of the highly performed statistics are not sufficiently explained. The important findings of this study are the fact that basic neonatal care is mainly absent in the four major maternity wards of Nepal and that there is a tremendous need for training of neonatal care algorithms and skills.

Response. We used advanced statistics to explain the phenomenon of mistreatment to newborns. We have explained the statistics performed in page 13, line 10-22. We agree than there is a need to improve the clinical care process in four hospitals of Nepal and have now added it in page 20, lines 7-9 as “Health workers will require training to improve their skills to inform women and consent for care being provided to newborn infants.”

---

## [Editor Report · Decision Letter 2]

18 Jan 2021

Mistreatment of newborns after childbirth in health facilities in Nepal: results from a prospective cohort observational study

PONE-D-20-21414R2

Dear Dr. KC,

We’re pleased to inform you that your manuscript has been judged scientifically suitable for publication and will be formally accepted for publication once it meets all outstanding technical requirements.

Kind regards,

David Desseauve, MD, MPH, PhD

Academic Editor

PLOS ONE

---

## [Editor Report · Acceptance letter]

26 Jan 2021

PONE-D-20-21414R2 

Mistreatment of newborns after childbirth in health facilities in Nepal: results from a prospective cohort observational study 

Dear Dr. KC:

I'm pleased to inform you that your manuscript has been deemed suitable for publication in PLOS ONE. Congratulations! Your manuscript is now with our production department. 

Kind regards, 

on behalf of

Dr. David Desseauve 

Academic Editor

PLOS ONE